# Physiological and Productive Responses of Two *Vitis vinifera* L. Cultivars across Three Sites in Central-South Italy

Filippo Ferlito [1], Elisabetta Nicolosi [2,*], Angelo Sicilia [2], Clizia Villano [3], Riccardo Aversano [3] and Angela Roberta Lo Piero [2]

1    Council for Agricultural Research and Economics, Research Centre for Olive, Fruit and Citrus Crops, 95024 Acireale, Italy; filippo.ferlito@crea.gov.it
2    Department of Agriculture, Food and Environment, University of Catania, 95123 Catania, Italy; angelo.sicilia@unict.it (A.S.); rlopiero@unict.it (A.R.L.P.)
3    Department of Agricultural Sciences, University of Naples Federico II, 80055 Naples, Italy; clizia.villano@unina.it (C.V.); raversan@unina.it (R.A.)
*    Correspondence: elisabetta.nicolosi@unict.it

**Abstract:** Grapevine adaptation to drought involves morphological, anatomical, and physiological modifications that could be viewed as a measure of drought avoidance. The main vine responses to drought consist of the regulation of carbon assimilation as a consequence of limited stomatal conductance, which is reflected in changes in plant water status. In this factorial study (2020–2021 growing seasons), two red cultivars, the local 'Aglianico', widely grown in Central-South Italy, and the international 'Cabernet Sauvignon', were used to evaluate how their interaction in three different environments can modify physiological adaptations and how yields and their qualitative traits can be modified. The lowest leaf water potential (−0.68 Mpa) for the two cultivars was registered in Molise, while three most stressed vine was found in Sicily for Aglianico (−1.86 MPa). At least in two of three locations, Molise and Campania, the detected stomatal conductance and the leaf water potential have shown that Cabernet Sauvignon can be classified as a near-isohydric cultivar, whereas Aglianico can be categorized as a near-anisohydric cultivar. The interactions between genotype x environment highlight different levels of adaptability between the two cultivars in different sites during each season. The data presented here contribute to a better understanding of the effects of genotype and environment interactions in progressive dry cultivation and how these interactions can modify the qualitative traits of grapes.

**Keywords:** climate change; Aglianico; Cabernet Sauvignon; transpiration rate; stomatal conductance; isohydric; anisohydric

## 1. Introduction

In 2021, grapevine (*Vitis vinifera* L.) was cultivated in 6.9 Mha, producing 78 Mt worldwide [1]. One of the most important areas, both for its long history of cultivation and the high qualitative traits of the produced wines, is the Mediterranean basin, in which the grapevine is cultivated in 3.2 Mha (47% of the total), producing 29.2 Mt (38% of the total). In this area, in 2020, more than 130 MhL of wine was produced, 51% of the worldwide production. Currently, a consistency of 5000–10,000 cultivars has been reported for grapevine [2,3]. This large genetic platform was derived from grape hybridization, breeding, and selection of somatic mutations, which started with the domestication of the species [4,5]. For each region, most of the well-known cultivars are closely related to their belonging territory due to their strong historical and cultural ties with the place of origin [6]. The productive traits are influenced by the connection between the genotypes and the environmental conditions, soils, and natural resources, and they can influence the wine characteristics [7,8]. The connection between a territory and its main cultivar, in particular the indication of origin, is one of the most considered elements taken into

account by consumers for their wine choice [9]. For traditional and ancient viticultural areas, the specifications of a denomination of origin partially carry the same information by establishing constraints on the grape varieties that can be used [10]. In contrast, in modern areas for viticulture, in countries in which there are no denomination systems, productions are obtained by a few phenotypically stable international genotypes with a lower level of typicality that can be accepted by consumers. However, in several typical grapevine territories, the introduction of new cultivars, called alloctonous or international, is common. These cultivars are well adapted to different pedoclimatic conditions, thus ensuring production with constant profiles [11–13]. Considering the proposed "climate change scenario", the risk for traditional viticulture is higher than that practiced in modern areas, mainly involving the substitution of low-plastic local cultivars. Moreover, the modification of the dedicated areas for viticulture, especially in Mediterranean areas, could be possible in the next decades as a consequence of an increase in average temperature with enhanced levels of UV-B incidence and a decrease in water availability [14,15]. In this area, the vines' response to drought and solar radiation can vary within cultivars [16]. Among the atmospheric variables, solar radiation, photosynthetically active radiation (PAR), and, in particular, the incident fraction on the leaf surface ($Q_{leaf}$) are taken into account as biophysical parameters useful to define how vines are related to a proper environment, and their variability is responsible for the modulation of gas exchanges [17]. $Q_{leaf}$ represents the main source of energy used by the vine for stomatal conductance, photosynthesis, and transpiration processes, and as a consequence, it also influences the leaf water potential, determining vine efficiency, which is also responsible for plant growth and productivity. Therefore, all these variables can vary according to the phenological stage and throughout the site [18,19]. Moreover, the abovementioned parameters, mainly photosynthesis and transpiration, are also related to carbon dioxide ($CO_2$) exposure and to the substomatal $CO_2$ concentration plus leaf size and anatomical characteristics; the latter may alter the internal $CO_2$ resistance from the substomatal cavities to the sites of carboxylation and consequently the photosynthetic rate [20].

Despite contradictory reports of cultivar-specific responses [21,22], grapevine adaptation to drought generally involves morphological and anatomical characteristics, such as alterations in leaf area [23], root/shoot ratio [24], xylem vessel size and conductivity [25], and physiological [26] and productive traits [27]. The main physiological vine response to drought consists of the regulation of carbon assimilation as a consequence of limited stomatal conductance, thus increasing water use efficiency [28]. In arid environments, the regulation of stomatal opening represents the most important mechanism for the plant control of water loss, and it has long been known that genetic variation as a response to water deficits [28] is possible within crop species and cultivars, distinguishing iso- and anisohydric species and cultivars [16]. Therefore, these physiological parameters could be considered a measure of drought avoidance [29]. For example, the maximum transpiration rate and early stomatal closure have been proposed as traits of drought avoidance [30,31].

Depending on the vines' habitus, in particular on quickness and speed to the stress response, vine cultivars were sometimes described in terms of 'isohydric' (in which the control of the stomatal conductance is high and the hydric potential fluctuations in response to soil water deficit are minimized) vs. 'anisohydric' (in which a lower control over stomatal aperture resulting in substantial reductions in leaf water potential with increasing soil water deficits) [32]. However, the physiological variables may be modulated by either the environment or changes in soil moisture [33] and can have a role in productivity and berry and wine quality when environmental conditions are limiting [34]. For example, a strong reduction in plant carbon assimilation due to a severe decline in photosynthesis, as well as a partial loss of canopy leaf area, can determine a reduction in wine production [35].

It is well known that an excessive water deficit in post-veraison reduces leaf photosynthesis and thus impairs sugar accumulation. Concordantly, as the yield qualitative traits indicated [36], a lower content of soluble solids in highly stressful environments was observed. On the contrary, the environment strongly influences the grape acidity, especially

considering the higher accumulation of organic acids at high latitudes [8], while it seems to play a less important role in the pH expression [37]. The growing environment influences the expression of grapevine genes involved in the catabolism of key aroma compounds, which are enhanced when water deficit occurs in pre-veraison, while no effect is detected for post-veraison stress [38]. Since flavonols act in grapes as UV- and photo-protectors, their concentrations turned out to be higher in more exposed to sunlight sites [39]. The effect of different pedoclimatic conditions on berries' qualitative traits is due to differences in the microbiota composition of berries. In fact, the microbiota of berries varies according to geographical location, and that variation can be a factor that potentially contributes to regional wine characteristics [40].

This study is part of complex research on the climate change scenario based on a multidisciplinary approach. The general aim of the research is to highlight that the maintenance of grapevine production will require adaptation strategies to climate change under rainfed conditions in one of the most important areas for viticulture. In particular, two black cultivars, 'Aglianico', widely grown in southern Italy (mainly in the Campania and Basilicata regions), and the ubiquitous 'Cabernet Sauvignon' international cultivar, were used to evaluate the interaction among the genotypes and three different environments that can modify or limit the physiological conditions of adaptation. The data presented here may be of great relevance in evaluating the interactions between genotype and environment in progressive dry cultivation, especially when environmental stress, simulating a climate change scenario, can limit the quality standards for premium wine.

## 2. Material and Methods

### 2.1. Site Description, Plant Material, and Experimental Design

A field factorial study was conducted during two consecutive years (2020 and 2021) in six 10-year-old commercial *Vitis vinifera* L. vineyards cultivated with the Aglianico and Cabernet Sauvignon black cultivars in three Italian regions: Molise (San Biase, Campobasso district Aglianico: 41°72′49.57″ N, 14°57′28.16″ E; Cabernet Sauvignon: 41°71′49.35″ N, 14°57′22.93″; 600 m a.s.l.), Campania (Galluccio, Caserta district; Aglianico: 41°33′38.27″ N, 13°89′31.16 E; Cabernet Sauvignon: 41°33′32.79″, 13°90′39.65″ E; 125 m a.s.l.) and Sicily (Zafferana Etnea, Catania, Aglianico: 37°39′35.61″ N, 15°05′08.59″ E; Cabernet Sauvignon: 37°41′18.66″ N, 15°07′28.42″ E; 720 m a.s.l.). All vines were grafted onto 140 Ruggieri rootstocks in North–South rows on the slopes of the hills in Molise and Campania or mountains in Sicily. In Molise (vines spaced 1.20 m × 2.90 m for both cultivars) and Campania (Aglianico vines spaced 1.50 m × 2.90 m; Cabernet Sauvignon: vine spaced 1.0 m × 2.70 m), vines were simple Guyot trained, with a formation height of 60 cm. In Sicily, (Aglianico vines spaced 1.10 m × 1.10 m; Cabernet Sauvignon: 1.10 m × 1.30 m) vines were bush trained at 0.5 m with two to six main branches, each branch was spur-pruned to one spur, with two buds per spur. The vineyards were tilled and rain-fed. The study was a completely randomized design with three independent plots of five rows, each containing 30 vines. All measurements were made on seven 'index' vines per block. For Aglianico, three sites and two years were considered; meanwhile, the Campania site was missed in the first season for Cabernet Sauvignon.

### 2.2. Climate and Soil

Daily temperature and rainfall data were provided by in situ climatic stations [23] (Figure S1). For each year, monthly minimum, mean, and maximum air temperatures, rainfall, reference evapotranspiration, cultural evapotranspiration, and deficit leaf air vapor pressure were registered in the experimental vineyards. The chill hours (from 1 October to 28 February) and the growing degree days, calculated from local meteorological data for the years 2020–2021, were 1560 and 1650 in Molise, 840 and 2112 in Campania, and 580 and 2045 in Sicily. The soil moisture and temperature (averaged along soil profiles) from 1 April to 31 October were 55% and 12.8 °C, 41% and 17.8 °C, and 15% and 16.5 °C in

Molise, Campania, and Sicily, respectively. Soils were analyzed [23] (Table S1) and classified according to the USDA [41–43].

### 2.3. Treatments

The combined 6 treatments analyzed were (1) Aglianico-Molise, (2) Cabernet Sauvignon-Molise, (3) Aglianico-Campania, (4) Cabernet Sauvignon-Campania, (5) Aglianico-Sicilia, and (6) Cabernet Sauvignon-Sicilia.

### 2.4. Physiological Behavior and Vine Water Status Measurements

For the vines' physiological behavior monitoring, the observations were made according to the Biologische Bundesanstalt, Bundessortenamt and CHemical industry (BBCH) [44] at the principal growth stage 6: Flowering (BBCH69 End of flowering), at the principal growth stage 7: Development of fruits (BBCH75 Pea-sized berries), at the principal growth stage 8: Ripening of berries (BBCH85 Softening of berries) both years and on the same day of the year and at principal growth stage 5: Inflorescence emerges (BBCH57 Flowers separating) during the second season.

Two leaves per vine were collected for leaf water potential measurements ($\Psi_L$), one from the main shoot and one from lateral shoots (42 leaves in total). The measurements were taken on fully expanded leaves using a Schöelander pressure chamber [45]. Leaf gas exchange, namely, net photosynthesis (A, mol $CO_2$ m$^{-2}$ s$^{-1}$), transpiration rate (E, mmol $CO_2$ m$^{-2}$ s$^{-1}$), and stomatal conductance (gs, mol $CO_2$ m$^{-2}$ s$^{-1}$), was measured on well-exposed, mature main and lateral leaves (42 leaves in total) between 12:00 and 14:00 h, when solar radiation is at maximum intensity, using a portable IRGA (Leaf Chamber Analyser—ADC LCA4 Bio Scientific Ltd., Rickmansworth, England). Photosynthetically active radiation (PAR) incident on the leaf surface ($Q_{leaf}$, µmol m$^{-2}$ s$^{-1}$ of photons) was determined simultaneously with the measurements of the eco-physiological variables, using the sensor coupled to the porometer chamber, always exposed perpendicularly to incident sunlight on the leaf surface throughout each reading. Moreover, leaf temperature ($T_l$, °C) and $CO_2$ internal concentration (Ci, µmol mol$^{-1}$) were also measured [46,47]. Following [48], the photosynthesis/transpiration ratio was taken as an estimation of instantaneous water use efficiency (WUE), and the ratio between photosynthesis/stomatal conductance, which is known as intrinsic water use efficiency, was calculated.

### 2.5. Crop Yield and Berry Characteristics

In 2020 and 2021, yield per vine and grape composition were analyzed at harvest, which occurred at commercial ripening. For yield assessment, all bunches per vine and shoot were counted and weighed, and the total fresh weight yields per vine were recorded. Two bunches from each index vine (42 bunches) were randomly selected and dissected to determine bunch weight and mean berry weight. A 100-berry sample per experimental block was divided into three subsamples, crushed with a manual press, and free-run juice was utilized to determine the total soluble solids (TSS) with a digital refractometer with temperature correction (RX-5000 Atago Co., Ltd., Bellevue, WA, USA), must pH and titratable acidity (TA), using an automatic titrator (Titrino Model 798, Metrohm, Riverview, FL, USA) with 5.0 mL juice samples being titrated against 0.1 M NaOH to pH 8.2; total acidity (TA) was expressed as g/L of tartaric acid equivalents [27].

### 2.6. Statistical Analyses

Analysis of variance (ANOVA) was performed with Jamovi 2.0.0 statistical software (The Jamovi project, 2021). A two-way analysis of variance (ANOVA) was carried out for each cultivar and year on the differences among phenological stages, sites, and different leaves (main and later). A post hoc analysis based on the Tukey HSD test (Tukey Honest Significant Differences) was performed at significance levels (p-value) of 0.05, 0.01, and 0.001. The comparison among the three sites was carried out through the Principal Component Analysis (PCA) of the average data to summarize the specific responses of the two cultivars

graphically and to highlight possible similarities or dissimilarities implemented in R 4.0.3 statistical software [49].

Moreover, the physiological data used for the Pearson correlation analysis among all the collected parameters were computed using the psych package [50] implemented in R 4.0.3 statistical software [49]. The $\chi^2$ test of independence was computed with R 4.0.3 statistical software [49].

## 3. Results

### 3.1. Environmental Variables

The parameter photosynthetically active radiation (PAR) incident on the leaf surface ($Q_{leaf}$) ranged from 1957 (Cabernet Sauvignon, Molise, lateral leaf, flower separating) to 278 µmol m$^{-2}$ s$^{-1}$ (Aglianico, Campania, lateral leaf, berries pea-size). Regarding Aglianico, in 2020 in Sicily, at the first observation date and at the end of flowering, significantly lower values were observed for both the main and lateral leaves (Figure 1). At this stage, similar values were observed in Campania and Molise, representing the registered highest values. Later, at the pea-sized berries stage, no differences were observed in each site and for each type of leaf. At the last stage, softening of berries, the highest values were found in Campania, whereas in Molise, the $Q_{leaf}$ was the lowest. During the second year at the end of flowering, the highest $Q_{leaf}$ was found in Campania, and similar to the previous season, in Sicily, a significant low value was observed. During the softening of berries in Sicily, the highest values were observed, while in Molise and Campania, the incident $Q_{leaf}$ reduced its intensity. For Cabernet Sauvignon during 2020, significant differences were observed at the end of flowering and the softening of berries in Sicily compared to Molise (no data are available for Campania), while during the pea-sized berries stage, no differences were observed. For this cultivar, during the second year of research, a significantly lower value in $Q_{leaf}$ was observed at flower separating in Campania, at the end of flowering in Sicily, and at the pea-sized berries and the softening of berries stages in Molise. Additionally, for Cabernet Sauvignon, except in Molise early in the season, during the second year, the trend between the main and lateral leaves was similar.

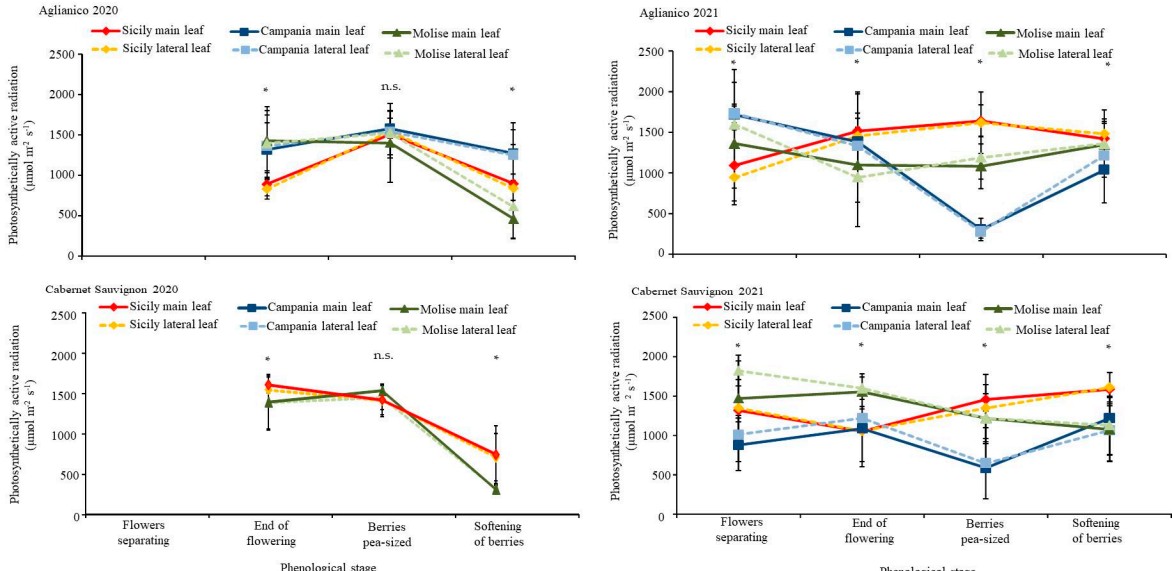

**Figure 1.** Photosynthetically active radiation (PAR) incident on the leaf surface ($Q_{leaf}$) was recorded on main and lateral leaves in two cultivars grown at three sites over two years. Measurements were made at the phenological stages BBCH57 (only in the second year) flowers separating, BBCH69 end of flowering, BBCH75 pea-sized berries, and BBCH85 softening of berries. Mean values for each phenological stage are reported (* = significantly different at $p \leq 0.05$, n.s. = not significant, based on Tukey's HSD test, bars indicate the standard deviation).

### 3.2. Vines Ecophysiological Response

The leaf water potential ranged from −0.68 (Mpa) both for the two cultivars in Molise, lateral leaf, flower separating, to −1.86 (MPa) for Aglianico in Sicily, main leaf at berries pea-size. The water status for Aglianico in 2020 showed differences at the end of the flowering and berry softening stages. During the first stage, a better performance (fewer negative values) was observed for the lateral leaves. At the berry softening stage, a significantly less negative value was recorded for the lateral leaves in Campania. In the second year, Aglianico showed the highest negative water potential values in Sicily up to the pea-sized berry stage, both for the main and lateral leaves. At the second and third stages in Campania, we observed less negative values. For Cabernet Sauvignon in 2020, a less negative water potential was recorded in Sicily at the end of flowering and the pea-sized berry stage for both main and lateral leaves. In 2021, a different behavior was observed between Sicily and Molise during flower separation and the end of flowering (Figure 2).

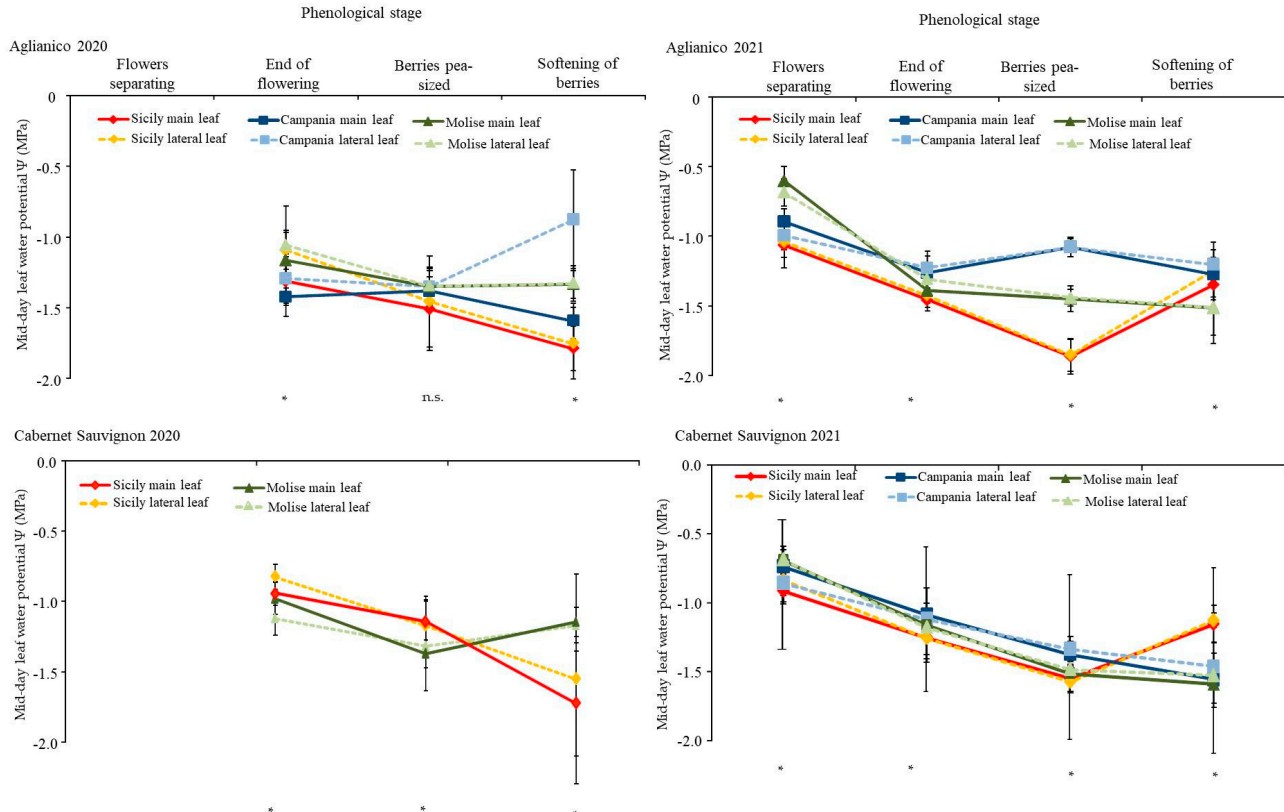

**Figure 2.** Mid-day leaf water potential was recorded on main and lateral leaves in two cultivars grown at three sites over two years. Measurements were made at the phenological stages BBCH57 (only in the second year) flowers separating, BBCH69 end of flowering, BBCH75 pea-sized berries, and BBCH85 softening of berries. Mean values for each phenological stage are reported (* = significantly different at $p \leq 0.05$, n.s. = not significant, based on Tukey's HSD test, bars indicate the standard deviation).

The substomatal $CO_2$ ranged from 82 (vpm) (Aglianico, Campania, main leaf, of flowering) to 310 (vpm) (Aglianico, Sicily, main leaf, berries pea-sized). The substomatal $CO_2$ concentration for Aglianico revealed a different performance of main and lateral leaves during the growth season and mainly during the second year. For Aglianico, lower values were registered at the end of flowering and the softening of berries in Sicily, while the highest contents were detected in Campania and Molise at the same stages. At the softening of berries stage, the opposite behavior was observed in Campania, in which the main leaves maintained a higher value in Molise, and no differences were observed among leaves. During the 2021 season in Sicily, the decline in $CO_2$ concentration in main leaves

was observed earliest at the end of flowering and was maintained until the pea-sized berries stage, while at the softening of berries in Sicily and Campania, the highest values were observed compared to the vines in Molise. The described trends observed for Aglianico were not observed for Cabernet Sauvignon in 2020. For this cultivar, in fact, no significant differences were registered for each site, year, and leaf except for low values observed for lateral leaves in Molise. Similarly, during the second year, the $CO_2$ concentration among sites was less variable; in fact, no significant differences were observed at the flower separating, at the end of flowering, and at the softening of berries, while a lower concentration in the main leaf in Sicily compared to the leaves (both main and lateral in Campania) was observed (Figure 3).

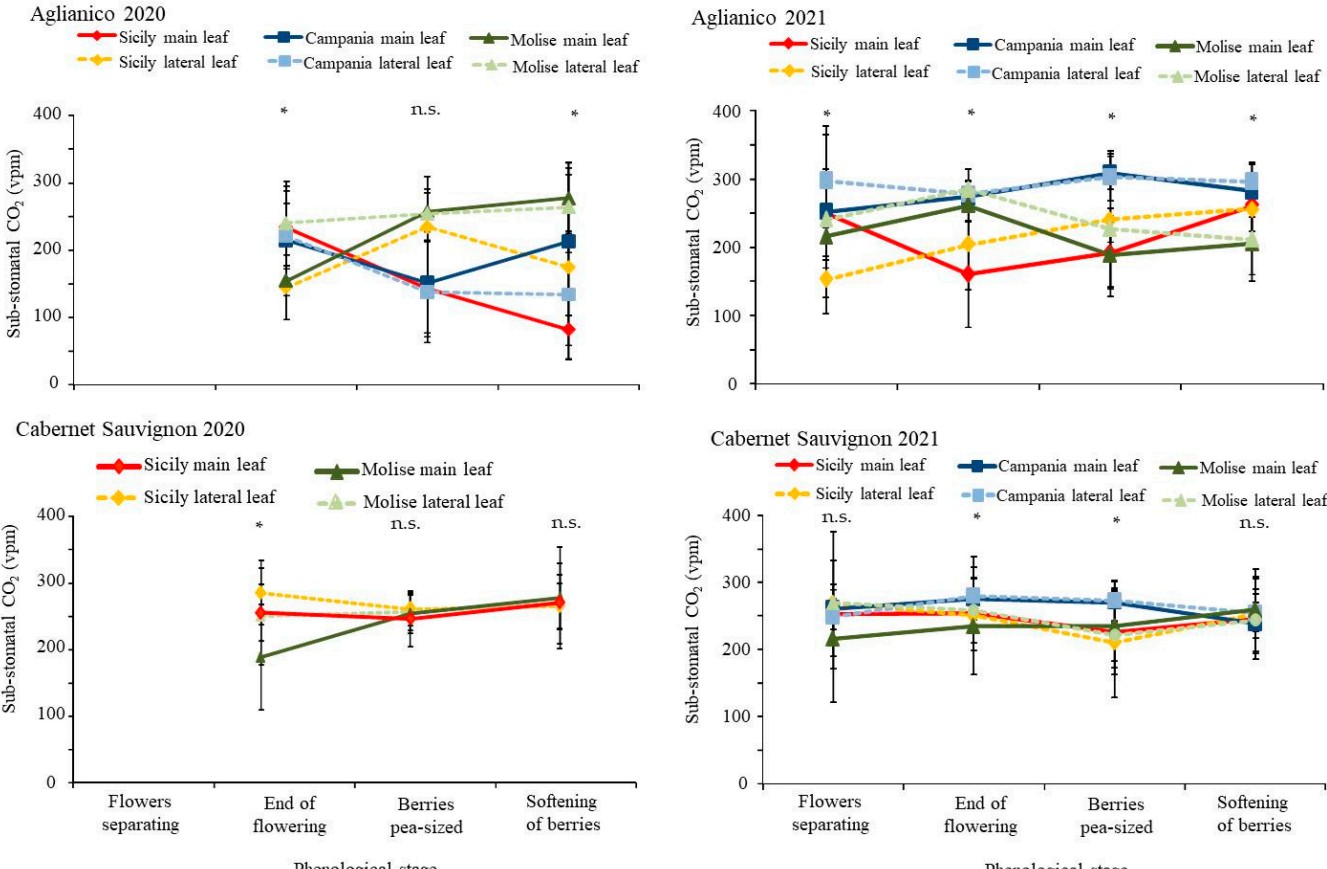

**Figure 3.** Sub-stomatal $CO_2$ recorded on main and lateral leaves was recorded in two cultivars grown at three sites over two years. Measurements were made at the phenological stages BBCH57 (only in the second year) flowers separating, BBCH69 end of flowering, BBCH75 pea-sized berries, and BBCH85 softening of berries. Mean values for each phenological stage are reported (* = significantly different at $p \leq 0.05$, n.s. = not significant, based on Tukey's HSD test, bars indicate the standard deviation).

The photosynthesis rate ranged from 2.08 ($\mu$mol m$^{-2}$ s$^{-1}$) (Aglianico, Campania, lateral leaf, softening of berries) to 13.87 ($\mu$mol m$^{-2}$ s$^{-1}$) (Cabernet Sauvignon, Molise, lateral leaf, end of flowering). The photosynthesis rate for Aglianico was higher in Molise for the main leaves at the first two stages, end of flowering and pea-sized berries, compared to all the other kinds of leaves and sites. During the second stage, in Campania, a significantly low efficiency in terms of photosynthesis rates was observed. At the softening of berries, the highest efficiency was observed for the lateral leaves in Molise. During the second year, the main leaves in Molise confirmed the highest efficiency, particularly at the first two stages and together with the laterals from the pea-sized berries stage to the softening of berries. At this stage, no differences were observed among all the treatments. During the mid-summer end of flowering and the pea-sized berries stage period in Sicily, the wrong

efficiency was registered for both main and lateral leaves. A strongly different behavior was observed for Cabernet Sauvignon. In Sicily, the main leaves during the first year were the most efficient, and the laterals were significantly highly efficient at the pea-sized berries stage and the softening of berries; they showed a significantly high value for the photosynthesis rate at flower separation and the softening of berries. Additionally, during the second year, the efficiency of the leaves of Cabernet Sauvignon in Sicily was similar to that of leaves in Molise. In particular, at the softening of berries stage, the highest values were observed in Sicily for both main and lateral leaves (Figure 4).

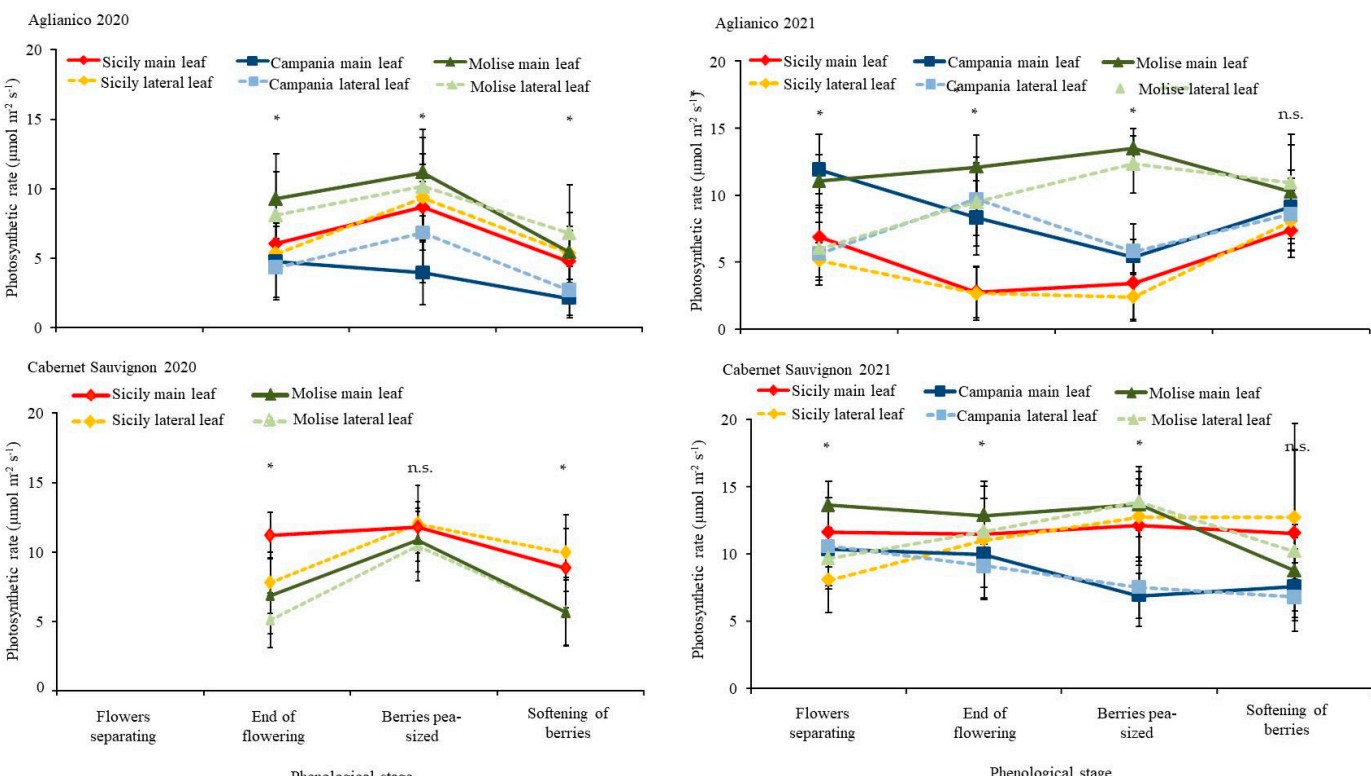

**Figure 4.** Photosynthetic rates recorded on main and lateral leaves were recorded in two cultivars grown at three sites over two years. Measurements were made at the phenological stages BBCH57 (only in the second year) flowers separating, BBCH69 end of flowering, BBCH75 pea-sized berries, and BBCH85 softening of berries. Mean values for each phenological stage are reported (* = significantly different at $p \leq 0.05$, n.s. = not significant, based on Tukey's HSD test, bars indicate the standard deviation).

The transpiration rate ranged from 0.76 (mmol m$^{-2}$ s$^{-1}$) (Aglianico, Campania, lateral leaf, softening of berries) to 8.15 (mmol m$^{-2}$ s$^{-1}$) (Cabernet Sauvignon, Molise, lateral leaf, end of flowering). Regarding the transpiration rate, for Aglianico in both years, the main differences were observed at the end of flowering. In particular, during the first year, the highest rates were reached in Molise, followed by Sicily, while in Campania, a lower amount of dissipated water was observed. In both years, the trends of the main and lateral leaves were similar at each site and for each date of observation. During the second year, at the end of flowering, the highest leaf transpiration rate was observed in Campania and Molise. During the pea-sized berries stage, the transpiration rate in Molise maintained the highest values. For Cabernet Sauvignon, the levels of transpiration were the highest at the end of flowering in Sicily, while the peak in Molise was reached later at the pea-size berries stage. In 2021, the highest levels at the end of flowering were registered in Molise, while from this step until the softening of berries stage, in Campania, the transpiration rates were lower than those at the other two sites. Additionally, for Cabernet Sauvignon, no significant differences were observed among the leaves from the main and lateral shoots (Figure 5).

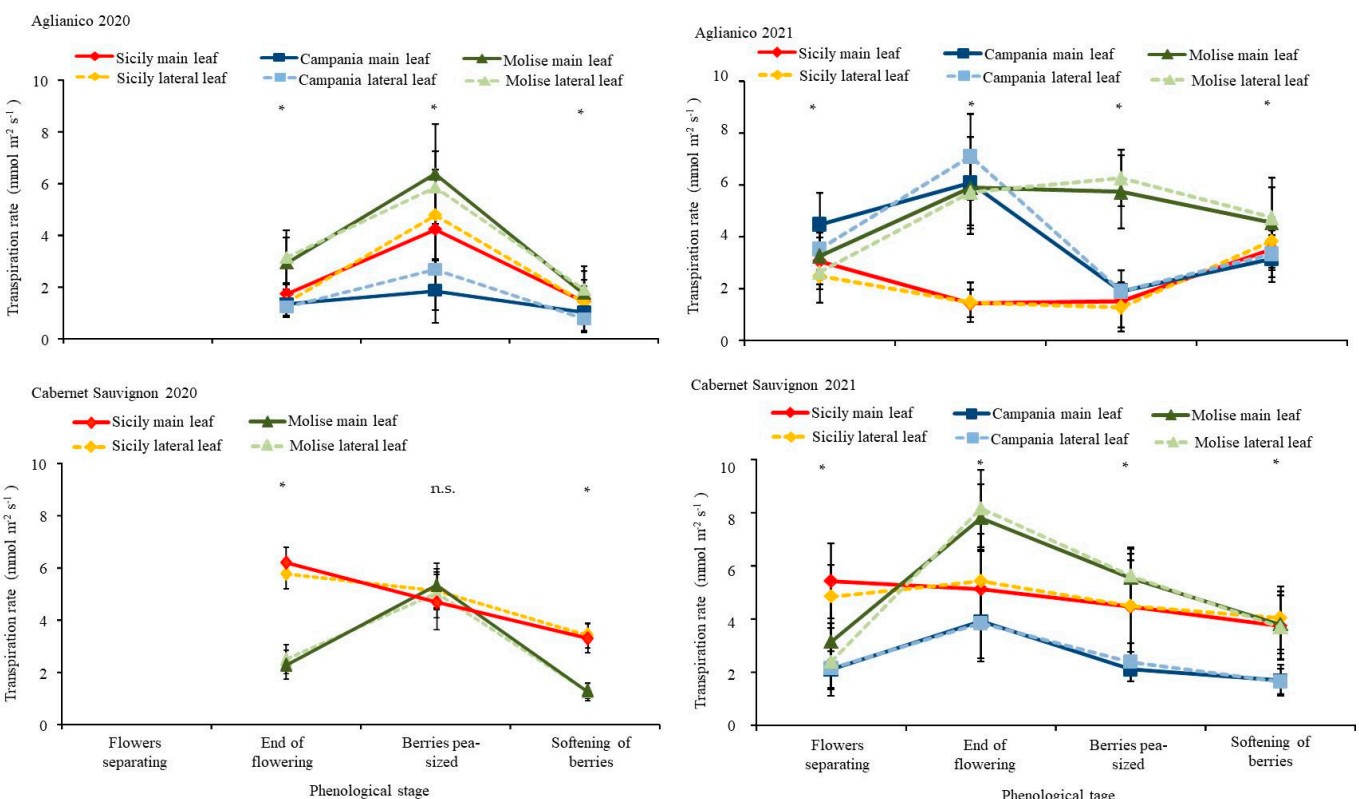

**Figure 5.** Transpiration rate recorded on main and lateral leaves was recorded in two cultivars grown at three sites over two years. Measurements were made at the phenological stages BBCH57 (only in the second year) flowers separating, BBCH69 end of flowering, BBCH75 pea-sized berries, and BBCH85 softening of berries. Mean values for each phenological stage are reported (* = significantly different at $p \leq 0.05$, n.s. = not significant, based on Tukey's HSD test, bars indicate the standard deviation).

The stomatal conductance ranged from 0,01 (mol m$^{-2}$ s$^{-1}$) (Aglianico, Sicilia, both main and lateral leaf, berries pea-size) to 0.6 (mol m$^{-2}$ s$^{-1}$) (Aglianico, Molise, lateral leaf, berries pea-size). Similarly, for this site, the stomatal conductance registered significant differences for Aglianico, and in this case, the highest values registered in Sicily at the end of the growth season were those in mid-summer. For this region, from the end of flowering to the pea-size berries stage, the performance of the lateral leaves was the highest. Additionally, for this parameter, no differences were observed for Cabernet Sauvignon (Figure 6).

Regarding the instantaneous water use efficiency (Table 1), the opposite behavior was detected for Aglianico between the two years of the research. In fact, while during 2020, only a significant difference was observed among the main and lateral leaves in Campania, in 2021, significant differences were observed early in the growth season between Molise (both lateral and main leaves), Sicily (main leaves), and Campania (lateral leaves). The main leaves in Molise maintained the highest values for the growth season. For Cabernet Sauvignon, the best performance at the first date was observed for lateral leaves in Molise. At the end of the season in Sicily, both lateral and main leaves showed a lower efficiency compared to Molise. During the second year, the best efficiency was confirmed in Molise, but in this case, the main leaves showed the best efficiency. Later, in Campania, the highest water use efficiency values were registered. At the softening of berries stage, a detrimental behavior was observed in Molise for each kind of leaf.

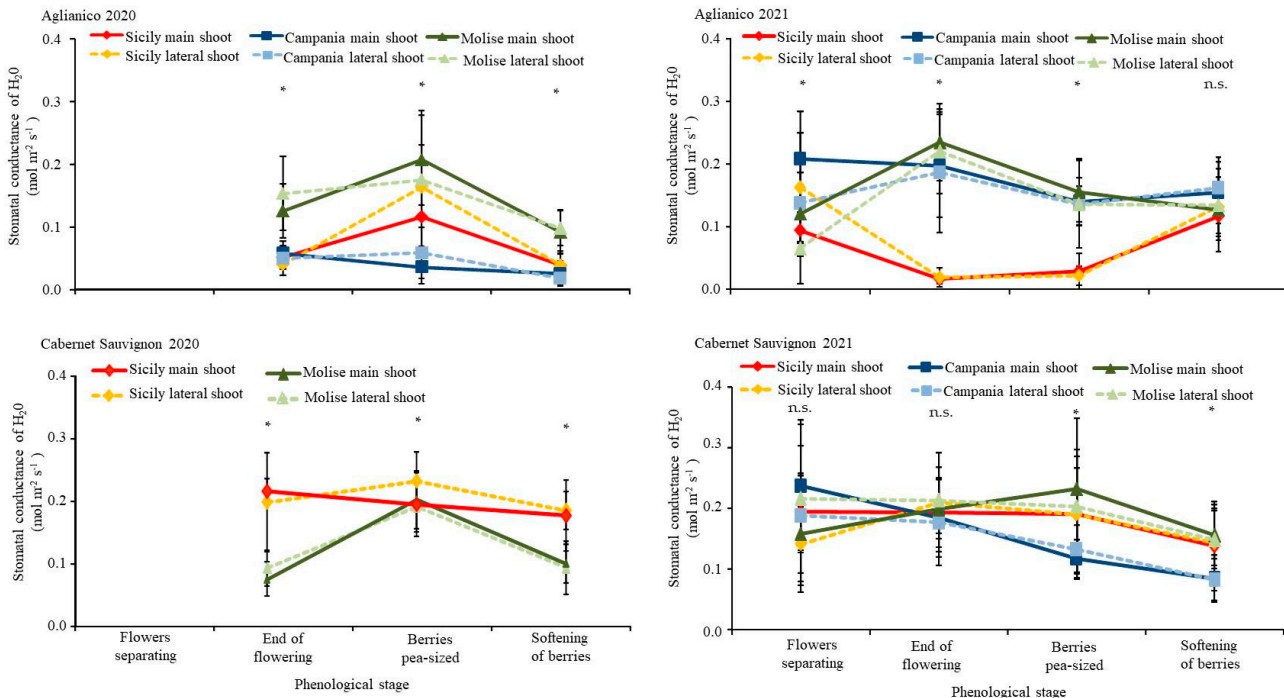

**Figure 6.** Stomatal conductance of $CO_2$ recorded on main and lateral leaves was recorded in two cultivars grown at three sites over two years. Measurements were made at the phenological stages BBCH57 (only in the second year) flowers separating, BBCH69 end of flowering, BBCH75 pea-sized berries, and BBCH85 softening of berries. Mean values for each phenological stage are reported (* = significantly different at $p \leq 0.05$, n.s. = not significant, based on Tukey's HSD test, bars indicate the standard deviation).

**Table 1.** Instantaneous water use efficiency [WUE] ($\mu$mol mol$^{-1}$) was recorded in two cultivars grown at three sites over two years. Measurements were made at the phenological stages BBCH57 (only in the second year) flowers separating, BBCH69 end of flowering, BBCH75 pea-sized berries, and BBCH85 softening of berries. Mean values for each phenological stage and year indicated by different letters are significantly different ($p \leq 0.05$, n.s. not significant, $\pm$ indicates standard deviation), based on Tukey's HSD test within years and cultivars.

| Year | Cultivar | Site | Leaf | Instantaneous Water Use Efficiency [WUE] ($\mu$mol mol$^{-1}$) | | | |
|---|---|---|---|---|---|---|---|
| | | | | Flowers Separating | End of Flowering | Berries Pea-Sized | Berries Softening |
| 2020 | Aglianico | Sicilia | Main | | 3.68 ± 0.97 [n.s.] | 2.3 ± 3.11 [n.s.] | 4.15 ± 2.21 [ab] |
| | | | Lateral | | 3.88 ± 0.77 [n.s.] | 2.10 ± 0.90 [n.s.] | 4.73 ± 2.05 [ab] |
| | | Campania | Main | | 3.87 ± 2.43 [n.s.] | 2.37 ± 0.90 [n.s.] | 2.54 ± 2.28 [b] |
| | | | Lateral | | 3.50 ± 1.49 [n.s.] | 2.80 ± 1.07 [n.s.] | 6.30 ± 8.03 [a] |
| | | Molise | Main | | 3.82 ± 2.80 [n.s.] | 1.85 ± 0.55 [n.s.] | 3.40 ± 1.54 [ab] |
| | | | Lateral | | 3.81 ± 6.15 [n.s.] | 1.80 ± 0.72 [n.s.] | 3.74 ± 1.54 [ab] |
| | Cabernet Sauvignon | Sicilia | Main | | 1.76 ± 0.27 [bc] | 2.63 ± 0.85 [a] | 2.70 ± 0.81 [b] |
| | | | Lateral | | 1.29 ± 0.38 [c] | 2.34 ± 0.57 [ab] | 2.91 ± 0.65 [b] |
| | | Campania | Main | | | | |
| | | | Lateral | | | | |
| | | Molise | Main | | 2.27 ± 1.36 [b] | 2.11 ± 0.71 [ab] | 5.25 ± 1.93 [a] |
| | | | Lateral | | 3.26 ± 1.53 [a] | 2.04 ± 0.49 [b] | 5.06 ± 2.25 [a] |

**Table 1.** *Cont.*

| Year | Cultivar | Site | Leaf | Instantaneous Water Use Efficiency [WUE] ($\mu$mol mol$^{-1}$) | | | |
| | | | | Flowers Separating | End of Flowering | Berries Pea-Sized | Berries Softening |
|---|---|---|---|---|---|---|---|
| 2021 | Aglianico | Sicilia | Main | 2.37 $\pm$ 1.25 [c] | 1.99 $\pm$ 1.09 [b] | 2.46 $\pm$ 1.44 [c] | 2.19 $\pm$ 0.81 [n.s.] |
| | | | Lateral | 2.22 $\pm$ 1.07 [ab] | 1.75 $\pm$ 0.96 [ab] | 2.27 2.32 [ab] | 2.22 0.76 [n.s.] |
| | | Campania | Main | 2.88 $\pm$ 1.23 [ab] | 1.37 $\pm$ 0.30 [c] | 2.89 $\pm$ 0.80 [ab] | 3.02 $\pm$ 0.93 [n.s.] |
| | | | Lateral | 1.74 $\pm$ 1.11 [bc] | 1.36 $\pm$ 0.24 [c] | 3.07 $\pm$ 1.16 [a] | 2.71 $\pm$ 0.99 [n.s.] |
| | | Molise | Main | 3.62 $\pm$ 1.16 [a] | 2.11 $\pm$ 0.42 [a] | 2.47 $\pm$ 0.54 [ab] | 2.41 $\pm$ 0.84 [n.s.] |
| | | | Lateral | 3.65 4.49 [a] | 1.68 $\pm$ 0.44 [bc] | 1.98 $\pm$ 0.27 [b] | 2.37 $\pm$ 0.80 [n.s.] |
| | Cabernet Sauvignon | Sicilia | Main | 2.20 $\pm$ 0.50 [b] | 2.32 $\pm$ 0.45 [ab] | 3.26 $\pm$ 2.14 [n.s.] | 3.99 $\pm$ 1.20 [a] |
| | | | Lateral | 1.72 $\pm$ 0.57 [b] | 2.22 $\pm$ 1.07 [ab] | 3.45 $\pm$ 2.45 [n.s.] | 3.45 $\pm$ 2.55 [abc] |
| | | Campania | Main | 5.30 $\pm$ 1.99 [a] | 2.82 $\pm$ 0.94 [a] | 3.46 $\pm$ 1.34 [n.s.] | 4.55 $\pm$ 1.14 [a] |
| | | | Lateral | 5.54 $\pm$ 2.71 [a] | 2.57 $\pm$ 0.86 [a] | 3.24 $\pm$ 1.01 [n.s.] | 4.19 $\pm$ 1.32 [ab] |
| | | Molise | Main | 4.49 $\pm$ 0.97 [a] | 1.68 $\pm$ 0.35 [bc] | 2.63 $\pm$ 0.95 [n.s.] | 2.42 $\pm$ 0.75 [c] |
| | | | Lateral | 4.39 $\pm$ 1.86 [a] | 1.46 $\pm$ 0.50 [c] | 2.52 $\pm$ 0.99 [n.s.] | 2.87 $\pm$ 0.99 [c] |

The instantaneous intrinsic water use efficiency showed clear uniformity between the main and lateral leaves at each site for both cultivars and the date of observation. For Aglianico, the main differences were observed during the end of flowering between Molise and Campania. In the first site, a low efficiency was observed at the softening of berries stage. At the same stage, during the second year, no differences were observed. In contrast, early at the end of flowering and at the pea-size berries stage, the best efficiency was detected in Sicily. For Cabernet Sauvignon, the differences between sites and dates of observation were very limited. In fact, during 2020, high efficiency for the main leaf in Molise was observed only at the end of the flowering stage, and this difference was confirmed during the second year. Later, the behavior of the leaves at each site and the date of observations were absent. Table S2 reports all of the significant differences. A PCA was carried out in order to evaluate the effects of the site on the vine's physiological response (Figure 7). For Aglianico, a clear separation was observed between Campania and Sicily, compared to Molise; this behavior was well defined during the first year. On the contrary, during the second year, only Sicily and Molise confirmed their separation trend, while Campania showed a separation profile similar to Molise's. In Sicily, the parameters at the end of flowering and at the pea-size berries stage during the second year appear in the negative quadrant of component 2. For Aglianico, component 1 accounted for 46.4% of the explained variance, while component 2 accounted for 21.02% of the explained variance. With regards to the PC1, the main contribution was attributed to transpiration rate, stomatal conductance, and photosynthetic rate. Concerning the PC2, the photosynthetically active radiation, mid-day leaf water potential, and instantaneous water use efficiency strongly contributed to explaining the variability.

A different separation was observed for Cabernet Sauvignon. In particular, for this cultivar, during the first year, just at the phenological stage of the end of flowering, Molise observed a separation into the negative quadrant, whereas, during the second year, the separation among the three regions was less clearly defined at the different phenological stages. For Cabernet Sauvignon, component 1 accounted for 49.3% of the explained variance, while component 2 accounted for 17.6% of the explained variance.

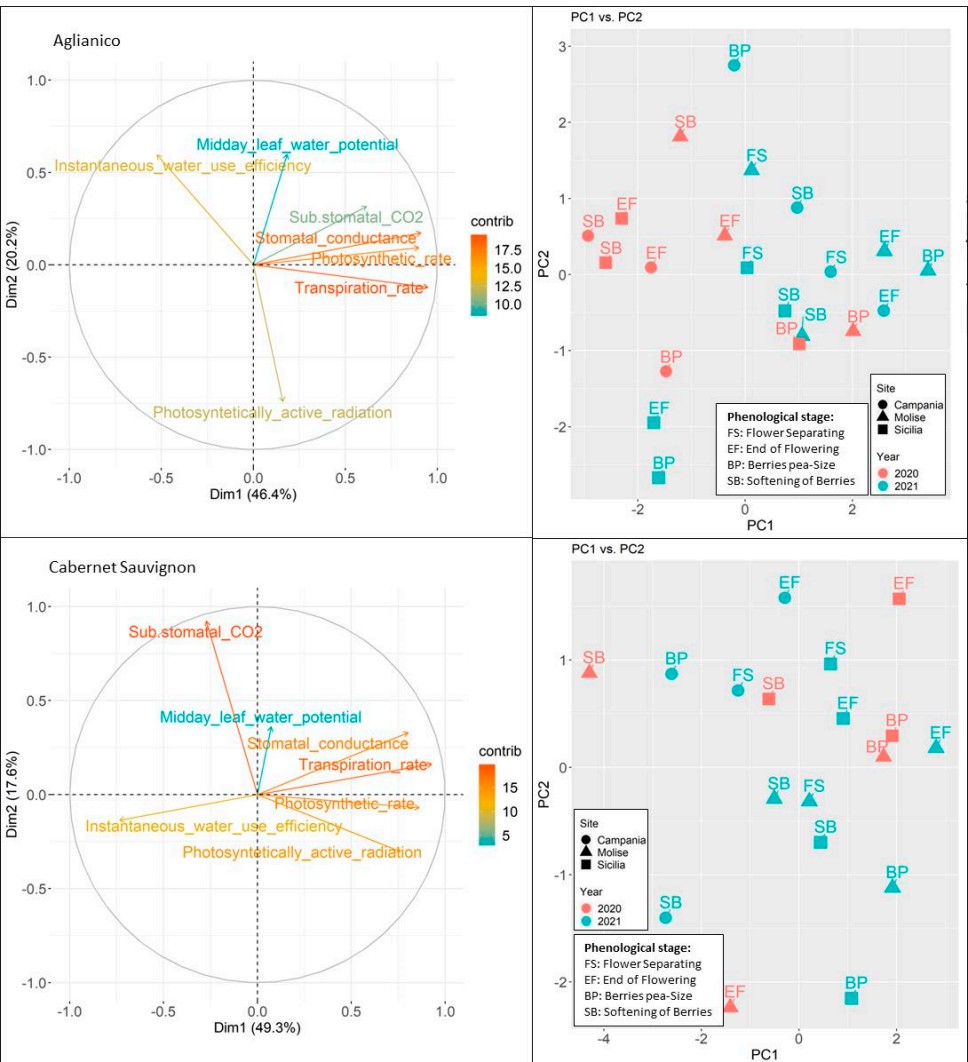

**Figure 7.** Biplot resulting from principal component analysis of the average data for Aglianico and Cabernet Sauvignon over two years (2020–2021) from three regions, Molise, Campania, and Sicily in southern Italy, as defined by the first two principal components. The cultivars are clustered on the basis of physiological indices at each phenological stage.

### 3.3. Yield and Its Components

The reproductive characteristics are reported in Table 2. AGL had the highest yield and bunch weight in MOL in both years. No differences were found for bunch length and berry number, while the berry and rachis weights were significantly lower in SIC in 2020 and 2021 (not shown). CAB showed a higher yield in MOL in 2021. Qualitative traits highlighted no significant differences in TSS for Aglianico in the three sites during the first year, while during the second year in Campania and Molise, a significantly higher content was detected compared to Sicily. However, in both years in Molise, the highest content of titratable acidity was registered. For Cabernet Sauvignon, TSS, pH, and TA were significantly higher in Molise than in Sicily during the first year. During the 2021 season, only TSS maintained the highest amount of TSS.

**Table 2.** Yield and its components in each of two cultivars, Aglianico and Cabernet Sauvignon, grown in three different sites, Molise, Campania, and Sicily. Measurements were made when all inflorescences were fully developed (Biologische Bundesanstalt, Bundessortenamt and Chemical industry: inflorescence 57). Mean values for each parameter and year indicated by different letters are significantly different (lowercase letter $p \leq 0.05$, uppercase letter $p \leq 0.001$, n.s. not significant, $\pm$ indicates standard deviation), based on Tukey's HSD test within years and cultivars.

| Parameter | Aglianico | | | | | |
| --- | --- | --- | --- | --- | --- | --- |
| | Molise | | Campania | | Sicily | |
| | 2020 | 2021 | 2020 | 2021 | 2020 | 2021 |
| Yield/vine (kg) | 6.80 ± 1.88 [A] | 9.10 ± 1.31 [a] | 2.11 ± 0.71 [B] | 6.24 ± 2.22 [b] | 0.24 ± 1.07 [B] | 0.44 ± 1.05 [b] |
| Bunch weight (g) | 558.60 ± 161.5 [A] | 345.80 ± 42.08 [n.s.] | 348 ± 56.35 [AB] | 298.20 ± 104.28 [n.s.] | 190 ± 15.41 B | 315.40 ± 85.23 [n.s.] |
| Berry weight (g) | 2.64 ± 0.22 [a] | 2.59 ± 0.22 [b] | 2.58 ± 0.17 [a] | 2.19 ± 0.27 [a] | 2.15 ± 0.03 [b] | 2.38 ± 0.19 [ab] |
| Total soluble solids (°Brix) | 21.59 ± 0.63 [n.s.] | 22.23 ± 0.88 [A] | 20.54 ± 0.55 [n.s.] | 23.23 ± 0.38 [A] | 21.06 ± 0.21 [n.s.] | 20.04 ± 0.73 [B] |
| pH | 3.48 ± 0.21 [b] | 3.34 ± 0.08 [B] | 3.85 ± 0.13 [a] | 3.63 ± 0.05 [A] | 3.5 ± 0.03 [ab] | 3.71 ± 0.13 [A] |
| Titratable acidity (g L$^{-1}$) | 10.40 ± 0.03 [A] | 17.1 ± 0.41 [A] | 9.80 ± 0.02 [A] | 8.00 ± 0.04 [B] | 5.30 ± 0.04 [B] | 7.6 ± 0.01 [B] |

| Parameter | Cabernet Sauvignon | | | | | |
| --- | --- | --- | --- | --- | --- | --- |
| | Molise | | Campania | | Sicily | |
| | 2020 | 2021 | 2020 | 2021 | 2020 | 2021 |
| Yield/vine (kg) | 4.84 ± 0.95 [A] | 8.57 ± 1.56 [A] | - | 4.96 ± 2.31 [B] | 2.45 ± 0.43 [B] | 2.79 ± 0.66 [B] |
| Bunch weight (g) | 352.60 ± 74.82 [a] | 444.6 ± 64.92 [n.s.] | - | 318.80 ± 102.27 [n.s.] | 258.40 ± 25.38 [b] | 346.60 ± 56.68 [n.s.] |
| Berry weight (g) | 1.97 ± 0.17 [n.s.] | 2.75 ± 0.18 [A] | - | 1.89 ± 0.22 [b] | 1.90 ± 0.12 [n.s.] | 1.50 ± 0.23 [c] |
| Total soluble solids (°Brix) | 22.33 ± 0.89 [A] | 24.45 ± 0.10 [A] | - | 23.00 ± 0.89 [AB] | 19.91 ± 0.04 [B] | 21.67 ± 0.98 [B] |
| pH | 4.01 ± 0.35 [A] | 3.80 ± 0.20 [n.s.] | - | 4.23 ± 0.37 [n.s.] | 3.61 ± 0.14 [B] | 3.74 ± 0.05 [n.s.] |
| Titratable acidity (g L$^{-1}$) | 9.8 ± 0.05 [A] | 7.7 ± 0.02 [n.s.] | - | 7.60 ± 0.01 [n.s.] | 5.1 ± 0.01 [B] | 7.5 ± 0.00 [n.s.] |

## 4. Discussion

The behavior observed for the two studied cultivars was certainly due to their genetic characteristics (G), to the pedo-climatic conditions in the three environments (E), and to the applied cultural practices (C). The interaction G × E × C allowed us to highlight different levels of adaptability for each genotype at the three sites in each of the ripening seasons. This result is a consequence of different performances in terms of vine growth and morphological traits [23], physiology, productivity, and yield components in commercial harvest data. During the present research, the fraction of PAR incident on the leaf surface ($Q_{leaf}$) highlighted different levels per site, season, and phenological stage. In Sicily, despite early in the season, this parameter was the lowest, and a significant increase in $Q_{leaf}$ highlighted a noteworthy climate trait during the pea-sized berry stage. It is important to note that from this stage until the harvest data, vines registered the smallest leaf area index and the most discontinuous canopies [23]. At this site, between the two cultivars, the sensitivity to PAR was less evident for Cabernet Sauvignon, in which the $Q_{leaf}$ increase was lower during the growth season. The measured reference $CO_2$ during our research was always below 450 vpm at each phenological stage and site and during the measurements of each kind of leaf for the two cultivars (not shown). This value is largely far from the threshold of 800 vpm, which is responsible for strong photosynthesis rate inhibition [51]. Similarly, those values below the 560 vpm threshold are also responsible for an overall decrease in air relative humidity and for an increase in the evaporative demand, which can determine a reduction in stomatal conductance of approximately 20% [16]. During our study for each site and phenological stage, the substomatal $CO_2$ never was above 300 vpm; therefore, the different physiological responses of the studied vines were driven by the other climatic variables and by their interactions with the genotypes plus the soil characteristics.

Analyzing the stomatal conductance and the leaf water potential detected in this research, it is possible to state that the two cultivars cannot fully fit into the isohydric group

or anisohydric group but could be classified as near-isohydric (Cabernet Sauvignon) and near-anisohydric (Aglianico). These classifications agree with the findings for Cabernet in relation to gs, photosynthesis, and WUE [50,51] and for Aglianico concerning the $\psi_l$ and E trends [52]. Moreover, it was suggested that iso/anisohydric behavior is influenced by the environmental conditions that plants are subjected to [53], seasonality [21,54] and soil moisture [33], and differences in leaf-to-air vapor pressure deficit (VPD) [55]. Thus, Cabernet Sauvignon was also categorized as near-anysohydric [56].

Accordingly, with the physiological response of the two cultivars combined with the ecological traits of the chosen sites, it is possible to consider Cabernet Sauvignon as a cultivar with a high stomatal conductance control that is able to maintain a high leaf water potential and its varietal characteristics independently from the growing conditions at almost two sites (Molise and Campania); for this cultivar, the leaf water potential was always maintained above or near −1.5 MPa except for the main leaf in Sicily during the first season. At this site, under those conditions, Cabernet Sauvignon showed near-anysohydric behavior. Therefore, the above-proposed classification (near iso- and near-anisohydric) better fits under mild and moderate water stress in Campania and Molise, while in Sicily, under high drought levels, vine stress can also occur during some periods of the growing season for this cultivar. From a practical point of view, Cabernet Sauvignon showed a high tolerance to limiting conditions, preserving soil moisture more efficiently than Aglianico. The latter was highly susceptible to stressful conditions that can occur at low latitudes in Campania or in Sicily. Aglianico showed several changes in stomatal conductance and leaf water potential. For this cultivar, the partial control of stomatal conductance was well shown for the main leaves in Sicily from the end of flowering to the berry pea-sized stage in mid-summer. In this region, despite a strong reduction in stomatal conductance, a lowering in water status was observed during the same period (−1.85 MPa). In contrast, in Molise and Campania, the leaf water potential was maintained at higher levels during the growth season (−1.5 MPa) as a near-isohydric cultivar. Additionally, the photosynthetic rate diminished for Aglianico during summer in Sicily, but the intrinsic water use efficiency (WUEintr) did not differ compared to other sites because hydric stress reduced the rate of photosynthesis but also the water loss, thus increasing the wage.

Our results were strengthened by the evidence of the different canopy expansions observed at each site and the morphological architecture of the leaves of the studied cultivars [23,57]. It was observed that a vine with a smaller canopy would use less water than one with a larger canopy, affecting grapevine water use [58–60]. Cabernet Sauvignon generally showed a higher compact canopy throughout Molise, Campania, and Sicily (in this site, the training system also contributed to reducing the canopy expansion) and smallest leaves (mainly for laterals) together with the surface that, at maturity, is highly sclerotic compared to the Aglianico leaves. Moreover, as described in previous research [23], the morphological response of Aglianico in the three sites highlighted a strong limitation in Sicily concerning shoot growth, and the basal mature leaves survived late in summer with strongly limited growth of laterals. Moreover, as already observed under field conditions [61], this expression of plant reactions to water stress seems to increase in sandy soils such as in Sicily. In contrast, a buffer effect on clay soils due to the higher capacity of this kind of soil to hold water and release it gradually to the plant was reported. Considering the climate change scenario and the registered vine physiological responses, it is possible to hypothesize the application of novel cultural strategies aiming to reduce the canopy volume, the evapotranspiration surface, and the total leaf area per vine. This trend makes it necessary to better understand the role and the response of different types of leaves to the canopy. Therefore, by clarifying the role of the lateral leaves and their behavior as reported in this study, in the future, it will be possible to adopt strategies to improve their growth when the efficiency of the main leaf is not guaranteed during all the growth season such as in Sicily or in Campania. On the contrary, in Molise, the lateral leaves seem responsible for the prolongation of the vegetative growth, thus representing a sink that could compete with bunches for the allocation of nutrients.

The Pearson's correlation matrix (Figures 8 and 9) allows a better comprehension of the relationships between the physiologically investigated variables. For Aglianico, the highest significantly positive correlations were found between the stomatal conductance and transpiration rate and between the photosynthesis rate and transpiration rate. Both correlations were found at each site. The correlation between the leaf water potential and the transpiration rate, although significant, showed the lowest values, and in Molise, it was negative. The relation between the stomatal conductance and the leaf water potential highlighted a strong correspondence and confirmed the anysohydric behavior of the cultivar, mainly in the drought regions in Sicily. The highest correlations between the photosynthesis rate and stomatal conductance were found in Sicily and Campania. Similar to the transpiration rate, the leaf water potential was less correlated with stomatal conductance and photosynthesis rate.

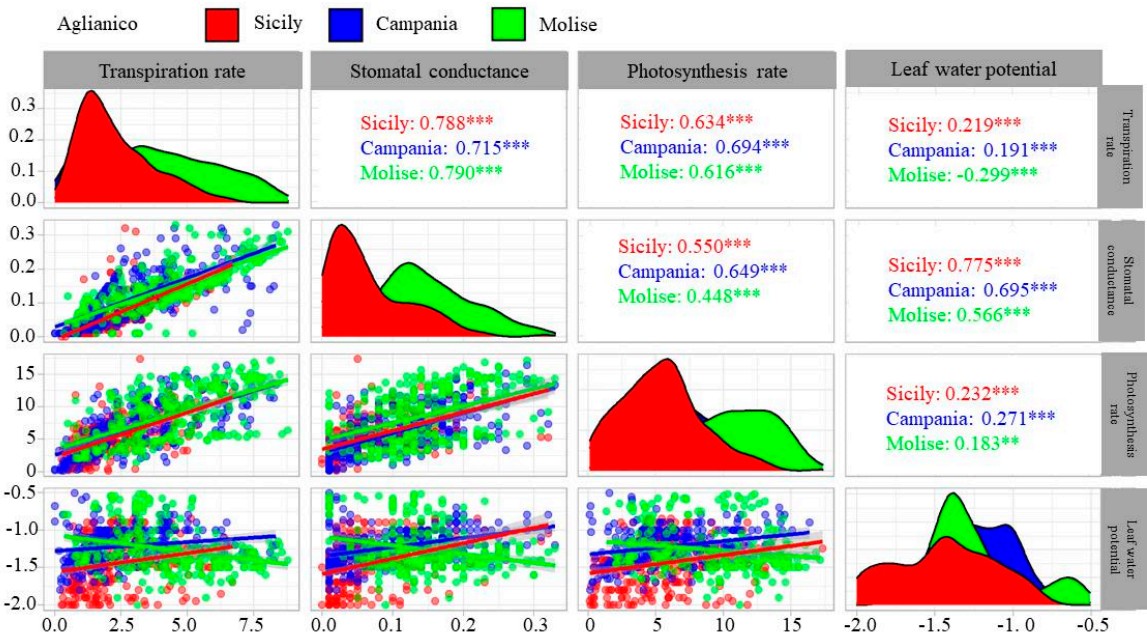

**Figure 8.** Pearson's correlation was registered in two cultivars grown at three sites in Aglianico over two years (** = significantly different at $p \leq 0.01$, *** = significantly dfferent at $p \leq 0.001$).

Additionally, for Cabernet Sauvignon, the correlation between stomatal conductance and transpiration rate was the highest at each site, while the correlation between the photosynthesis rate and transpiration was lower than the findings in Aglianico. Compared to the first cultivar, the correlation between leaf water potential and transpiration rate, stomatal conductance, and photosynthesis rate was low. Therefore, Cabernet Sauvignon showed substantial independence between these variables and, in particular, between stomatal conductance and leaf water potential. This correlation was negative in Molise but very low. Therefore, for both cultivars, the parameter leaf water potential seems less correlated with the other physiological parameters. In particular, the response in Molise, mainly for Cabernet Sauvignon, was not related to the physiological behavior of the vines. Compared to Aglianico, the leaf water potential and the stomatal conductance are not significantly correlated in Sicily.

The PCA analysis, mainly in 2021, confirmed the sensibility of Aglianico throughout a progressive drought environment (Molise < Campania < Sicily) in terms of reduction of photosynthetic rate and stomatal conductance at each phenological stage. Concerning the leaf water potential for this cultivar, the PC2 is more informative in both years, confirming that the most negative values were registered in Sicily rather than in Campania and Molise. For Cabernet Sauvignon, except for substomatal $CO_2$ and leaf water potential, all of the PC1 parameters highly contribute to explaining the observed variability.

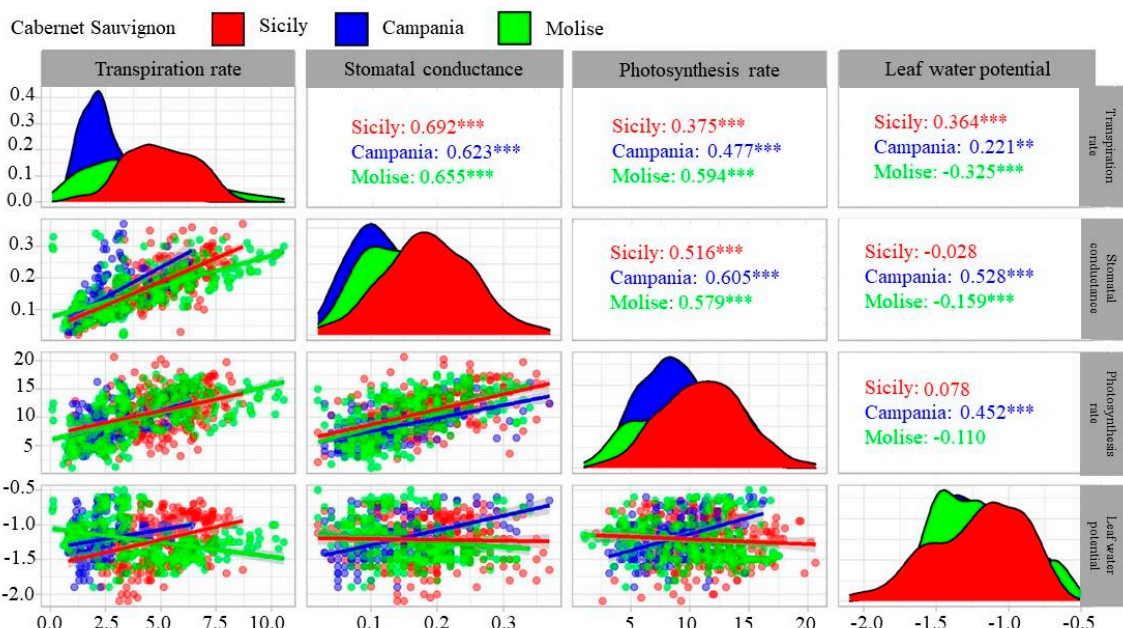

**Figure 9.** Pearson correlation matrix measured on Cabernet Sauvignon (\*\* = significantly different at $p \leq 0.01$, \*\*\* = significantly different at $p \leq 0.001$).

As regards the productive behavior, it seems it was influenced both by the bud load retained from the winter pruning and by the environmental conditions for quantitative and qualitative traits. The limited crop load for Aglianico in Sicily is also a consequence of a reduction in bud potential and observed fertility (not shown), as reported in previous research [23]. All these characteristics certainly influenced the yield quantity, particularly for Aglianico. This cultivar highlighted a significant yield contraction, which was not enough to allow this cultivar to reach satisfactory parameters in terms of quality, mainly TSS. However, the reduction in yield quantity and TA reduction allowed the wine to reach a better equilibrium between vegetative and reproductive growth, relocating the source/sink ratio [62]. This was not reached in Molise, where the TA level remained higher at harvest.

Studies conducted to compare the cultural responses to varying soil and climatic conditions, as in our case, will help to understand the adaptation limits of the territories currently planted with vines and the current terroirs. The research was realized in a very important area for viticulture in Italy where the traditional cultivar Aglianico represents a very important genetic resource, particularly for the Central South area of Italy. The evidence reported in the manuscript reveals that the pedoclimatic context on the Etna volcano slopes represents a particular site not perfectly suitable for its cultivation. This statement represents an indicator and a novelty element that should be considered when the climate change scenario is studied in viticulture.

## 5. Conclusions

The obtained results help to understand the interaction of genotype x environment and the different levels of adaptation to climate change, highlighting the importance of knowing the physiological plasticity of the genotypes.

At least in two of three locations, Molise and Campania, the stomatal conductance and the leaf water potential detected have shown that Cabernet Sauvignon can be classified as a near-isohydric cultivar, while Aglianico can be categorized as a near-anisohydric cultivar. The behavior of Cabernet Sauvignon was that of a cultivar with high stomatal conductance control that was able to maintain a less negative leaf water potential and its varietal characteristics, independent of the growing conditions at two sites, Molise and Campania. Aglianico was highly sensitive to stressful conditions that can occur at low latitudes, such as in Sicily. The data presented here may be of great relevance in evaluating the interactions

between genotypes and environments in progressive dry cultivation, especially when vine stress, simulating a climate change scenario, can limit the quality standards for premium wines. This study reports data that, integrated with those obtained by a multidisciplinary approach, can provide tools for a better and more comprehensive understanding of the behavior of cultivars under investigation.

**Supplementary Materials:** The following supporting information can be downloaded at: https://www.mdpi.com/article/10.3390/horticulturae9121321/s1. Table S1. Main physical and chemical parameters of soil for each study area: Molise, Campania and Sicily. Table S2. Physiological measurements recorded on main and lateral leaves recorded in two cultivars grown at three sites over two years. Measurements were made at the phenological stages BBCH57 (only in the second year) flowers separating, BBCH69 end of flowering, BBCH75 pea-sized berries, and BBCH85 softening of berries. Mean values for each parameter and phenological stage, indicated by different letters are significantly different ($p \leq 0.05$, n.s. not significant, $\pm$ indicates standard deviation), based on Tukey's HSD test within years and cultivars.

**Author Contributions:** Conceptualization, F.F., A.S., E.N. and A.R.L.P.; methodology, F.F., A.S. and E.N.; validation, E.N., F.F., A.S., C.V., R.A. and A.R.L.P.; investigation, E.N., F.F. and A.S.; data curation, F.F. and E.N.; writing—original draft preparation, F.F. and E.N.; writing—review and editing, E.N., F.F., C.V. and R.A.; and supervision, F.F., E.N., A.S. and A.R.L.P. All authors have read and agreed to the published version of the manuscript.

**Funding:** Progetti di ricerca di Rilevante Interesse Nazionale (PRIN2017) "Influence of Agro-climatic conditions on the microbiome and genetic expression of grapevines for the production of red wines: a multidisciplinary approach (ADAPT)", Ministero dell'Università e della Ricerca (MUR).

**Data Availability Statement:** Data are contained within the article and supplementary materials.

**Conflicts of Interest:** The authors declare no conflict of interest.

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
