# Peer review of "Physiological and Productive Responses of Two Vitis vinifera L. Cultivars across Three Sites in Central-South Italy"

_horticulturae, doi:10.3390/horticulturae9121321_

Round 1

Reviewer 1 Report

Comments and Suggestions for Authors

The manuscript title “Physiological and productive responses of two Vitis vinifera L. cultivars across three sites in Central-South Italy” is conducted in well manner but has significant mistakes that need to be addressed before making any decision. My major comments are as follows:

Reviewer Comments:

1-      In the start of abstract add 2-3 lines about the introduction/background of this study! Why you did this research! Raise some scientific questions that need to be addressed etc… In the end of abstract please add 2-3 concluding lines and outcomes about your research.

2-      Line 17-26 in abstract is copied and pasted at line 498-598 of conclusion section! Please rewrite the conclusion section, the outcomes of this research plus future research direction. Why you did this? Conclusion is not just copy paste of abstract.

3-      The rain, humidity, and temperature data should be given in graph form, added either in manuscript or as a supplementary material.

4-      Have you tested the soil Nutrients? Or chemical properties, because the soil properties also have a deep influence on physiological and productive responses of plants.

5-      Is there any level of significance applied to figures 1, 2, 3, 4, 5, and 6? Any significant difference among values? The figures show no signs of significant difference! Is all figures data is non-significant? Also, level of significance is not mentioned in the figure legends.

 6-      Line 452 correct the figg to fig.

Comments on the Quality of English Language

English doesn't have any major issues.

Author Response

Dear reviewer, please find in attach 

  • the revised manuscrip
  • the response to the specific comments
  • the supplementary material: Figure S1, tableS1, tableS2
  • English certificate

Best Regards

Reviewer 2 Report

Comments and Suggestions for Authors

Dear authors

The following modifications are required

Abstract

ü  In general, this section is poorly written. It is written simply. This section should include. As a result, this section should be improved.

ü  Before describing the goal, the authors must define the issue in a single line and explain why they chose this approach to study this review.

ü  No information about the type of experimental design and its component is available in this manuscript.

ü  Some quantitative data should be added

ü  In the final line of the abstract, the authors should present a decisive conclusion derived from the research and provide a single line of future prospects.

Keywords

ü  The content of keywords did not reflect the content of this manuscript and the words used for forming the title should not be used as the keywords. So, the structure of keywords should be changed.

Introduction

ü  Detail information about the impacts of different environmental factors on the biochemical parameters

ü  The authors should give some lines about the knowledge gap which their reviews have covered along with the hypothesis statement

ü  Also, the authors should provide a novelty statement at the end. What new things authors have done or correlated in this research compared to old ones?

ü  The general and specific aim should be specified

Materials and Methods

ü  No information about the type of experimental design and its component is available in this manuscript.

ü  The authors should mention the method of comparison between the means in the section of statistical data analysis

ü  The authors should write the number of replications and the number of plants per replications.

ü  The authors should mention the type of tissue used for the studied traits

ü  All measurements should be supported by the references

ü  The type of soil (Silt, sand….etc) should be mentioned

ü  All abbreviations should be written in full name

Results and discussion

ü  In general, the figures are not presented clearly.

ü  The status of significant of each studied parameter should be mentioned at the beginning of the text

ü  All figures should be subjected to the statistical analysis by adding the letters

ü  The authors should mention the scored data when they explain the maximum and minimum of the studied traits.

ü  All captions should be improved, showing the contents of tables and figures

ü  The method of the comparison of means (LSD, Duncan, Tukey, Dunnett) should be included

ü  A PCA plot should be created to better understand the studied traits.

ü  The correlation and probability values of each pair of traits should be mentioned in the text

ü  The discussion is weak. The authors should interpret all results obtained in this study by adding some information about the results obtained in their study. The authors should explain how all of the findings from this study relate to their own findings. The authors should explain the impact of environments on the studied traits by adding the mechanism of affecting

Conclusion

ü  The authors should summarize the most significant findings because they have written this section in an easy-to-read manner.

ü  Future works about this research should also include additional works

Comments on the Quality of English Language

Extensive correction is needed

Author Response

(The authors gave the same response as above.)

Reviewer 3 Report

Comments and Suggestions for Authors

The manuscript entitled "Physiological and productive responses of two Vitis vinifera L. cultivars across three sites in Central-South Italy", presents the results from an experiment in which two black cultivars were used to evaluate the interaction among the genotypes and three different environments. The manuscript fits within the scope of Horticulturae. However, this manuscript cannot be accepted for publication in the journal because it presents severe flaws concerning the experimental design and the analysis of results. Only 4 examples:

1)     Regarding to experimental device, the two varieties growing in each of the three locations should be side by side, under similar pedoclimate conditions. From the GPS coordinates presented (Lines 112-116) it can be seen that this did not happen. Amazingly, 2 of the 6 plots are in the Ionian Sea. Other aspects related to the training system differ between varieties in the same locations. This greatly compromises the quality of the results.

2) Climate and soil (topic 2.2). The authors should present these results more completely. Your description is not enough.

3)  Statistical analysis: With 2 varieties and 3 locations, a two-way analysis of variance is essential to analyze these results. The presentation of the results in the article is very simplistic and difficult to follow. For some results (e.g. leaf gas exchange) no statistical test is presented. For example, in the legend of Figure 1 it is mentioned “... based on Tukey’s HSD test within sites and years.” – I can’t see any of this in the respective graphics!

4) Throughout the text several scientific inaccuracies deserve a more rigorous reading.

Author Response

(The authors gave the same response as above.)

Round 2

Reviewer 1 Report

Comments and Suggestions for Authors

the authors did sufficient revisions, and the current form of manuscript can be accepted for publication.

Comments on the Quality of English Language

English is fine

Author Response

Thank you very much for all the suggestions you gave us to improve the manuscript

Reviewer 2 Report

Comments and Suggestions for Authors

The authors have been addressed all comments

Comments on the Quality of English Language

moderate correction is needed

Author Response

Thanks a lot for the suggestions you gave us and which we have implemented to improve the manuscript

Reviewer 3 Report

Comments and Suggestions for Authors

The authors addressed all the issues raised and improved the clarity and quality of the manuscript. Therefore, I recommend that the manuscript can be published in its current form.